# Advancements in Mass Rearing the Air Potato Beetle *Lilioceris cheni*

**DOI:** 10.3390/insects13010065

**Published:** 2022-01-06

**Authors:** Emily C. Kraus, Rosemary Murray, Cassandra Kelm, Ryan Poffenberger, Eric Rohrig, Kate Fairbanks

**Affiliations:** Florida Department of Agriculture and Consumer Services, Division of Plant Industry, Gainesville, FL 32608, USA; Rosemary.Murray@fdacs.gov (R.M.); Cassandra.kelm@fdacs.gov (C.K.); ryan.poffenberger@fdacs.gov (R.P.); Katherine.Fairbanks@fdacs.gov (K.F.)

**Keywords:** *Dioscorea bulbifera*, biological control, invasive species, Chrysomelidae

## Abstract

**Simple Summary:**

Mass rearing the air potato beetle, *Lilioceris cheni*, is a vital contribution to the integrated pest management of the invasive air potato vine. Here, the authors report on the production, distribution, and advancements of a mass rearing method. It was determined that adults are most successful on a diet of fresh air potato leaf. Although to reduce the amount of leaf tissue needed for overwintering populations, they can survive on artificial diet for several months and can be starved for several days to two weeks depending on previous diet. Larvae cannot survive continuously on artificial diet. This information allows those mass rearing beetles to reduce the amount of tissue grown at points in the annual cycle and shows that adult beetles can survive starvation while being distributed or for short periods after release. It also indicates that fresh air potato vine must be available in at least small quantities year-round. This information will assist those attempting to mass rear and distribute the beetle.

**Abstract:**

The air potato beetle, *Lilioceris cheni* Gressitt and Kimoto (Coleoptera:Chrysomelidae)*,* is a successful biological control agent of the air potato vine, *Dioscorea bulbifera* L. (Dioscoreales: Dioscoreaceae)*,* in the southern United States. *Lilioceris cheni* is currently being mass-reared by the Florida Department of Agriculture and Consumer Services Division of Plant Industry (FDACS-DPI) for biological control releases and research. The facility rears and releases over 50,000 adult beetles annually at approximately 1000 different locations. In addition to data on beetle production and distribution, studies on alternative larval and adult diets are described. Adults fed bulbils as the sole food source had reduced life spans compared with beetles given fresh air potato leaves. Adults survived without air potato leaves or bulbils for several days to two weeks depending on availability of leaves at emergence. Larvae did not survive on a modified artificial Colorado potato beetle diet containing fresh air potato vine leaves. Adults survived while consuming artificial diet but ceased oviposition. They, however, resumed egg laying less than one week after being returned to a diet of fresh air potato vine leaves.

## 1. Introduction

Air potato vine (*Dioscorea bulbifera* L.) is invasive to the United States and has spread throughout Florida, Alabama, Georgia, Hawaii, Louisiana, Mississippi, and Texas. There are many articles which describe the introduction and characteristics of this invasive species [1,2,3,4,5]. Traits such as rapid growth and high propagule pressure ultimately result in displacement of native species. Therefore, the plant must be managed but chemical and mechanical management methods have proven temporary and insufficient [6]. Chemical control is costly, requires repeated sprays over several years, and has non-target effects. Additionally, new vines often continue to sprout from underground tubers when herbicide treatments cease. Mechanical control of air potato is labor-intensive and time-consuming. Vines and bulbils can be hand-collected and destroyed but underground tubers are difficult to access and remove completely.

Biological control was considered the best management strategy for this invasive plant and has been shown subsequently to be cost effective [7]. The air potato beetle, *Lilioceris cheni* Gressitt and Kimoto (Coleoptera: Chrysomelidae) was identified as the most promising natural enemy. Host-specificity testing was performed on 41 non-target species, and *L. cheni* was found to be highly host-specific. Therefore, air potato vine must be grown as a food source for the mass rearing process. The vine does not sexually reproduce in the invaded range; therefore, aerial bulbils are the primary propagules [8]. Bulbils are either field collected from infested areas or grown along with the foliage in protected culture. The vine has underground storage organs known as tubers that sprout annually in the spring once suitable temperatures and irrigation are reached.

The air potato beetles collected from locations in Nepal and China in 2002 and 2011 represented two distinct biotypes and were released in Florida in 2012 [9,10,11]. The life cycle of the air potato beetle is holometabolous and spans 28 days, allowing for rapid production. Adults require approximately ten days post-emergence before they begin mating. During this time, they typically bite the air potato leaf veins resulting in cupping of the leaves. Eggs are most often laid in the cupped leaves. A female adult can lay more than 1200 eggs in her lifetime [10,12]. The pale white eggs become yellowish if fertilized, and larvae eclose in approximately four days. First instars are also yellowish but become grey to red with black legs as they progress through the four instars. The entire larval stage lasts approximately eight days. After this time, larvae drop off the plant and cluster in the soil to pupate. They secrete a whitish foamy substance that solidifies into a hard cocoon [12]. The pupal stage lasts approximately 14 days. Adults emerge and are assumed to live over five months under field conditions [9], and up to a year in the laboratory [13].

An efficient mass rearing protocol has been developed by the Florida Department of Agriculture and Consumer Services, Division of Plant Industry (FDACS-DPI) in Gainesville, FL, USA [14]. This protocol has consistently produced over 50,000 beetles annually for seven years and they have been distributed across the southeastern United States, including Florida, Georgia, Alabama, Mississippi, Louisiana, and Texas. It has been successful in providing beetles for research at several institutions, such as the University of Florida and Southern University and A&M college in Baton Rouge, Louisiana. The rearing methods can be upscaled or downscaled based on the size of the facility, staff, and quantity of beetles desired. As is the case in many mass rearing programs, host plant tissue is the major limiting factor. Therefore, the following experiments were conducted to determine the extent to which use of fresh air potato vine leaves can be minimized when mass rearing the beetles: (1) rearing beetle larvae and adults on artificial diet, and (2) rearing adults on air potato leaves versus bulbils.

## 2. Materials and Methods

### 2.1. Air Potato Vine Propagation

Host plant health is critical to the rearing process. A healthy, rapidly growing plant will produce more foliage and therefore support higher beetle production. Plants for beetle production and diet studies were grown indoors and outdoors according to the FDACS-DPI standard operating procedures [14].

Bulbils were field collected and spread out in a tray or directly on soil in a warm area (27 ± 2 °C). Blow-molded nursery pots (37–56 L) were half filled with a well-draining topsoil. Bulbils were placed on top of soil and pressed in slightly. Plants were watered, fertilized, and managed for pests according to the protocol [14]. The plants and tubers produced can grow in these containers for multiple years because tubers can remain in soil and continue to sprout vines under suitable conditions. Sprouting bulbils can be placed in the same containers as tubers from the previous year. The bulbils start rooting within two weeks.

Plants were allowed to grow for approximately two months before beetles were introduced for mass production. Leaf tissue was excised as needed for diet studies with a standard pruner. Care was taken to select leaves as near in size as possible. More than an ample amount of leaves and bulbils were supplied to beetles under experimental conditions. Bulbils for diet studies were field collected. In general, the bulbils selected were smooth, tan, and approximately the size of a tennis ball.

### 2.2. Lilioceris cheni Production and Releases

Beetles were mass-reared according to standard operating procedures at FDACS-DPI [14]. After collection from mass rearing areas, larvae were transferred to and maintained in 12-L plastic bins indoors at 27 ± 2 °C, 65 ± 10% RH, and 14L:10D. Indoor lighting was provided by 6500 K grow lights for diet experiments, only the Chinese biotype was used as there was an ample supply. Adult beetles were collected from multiple rearing areas for the mass rearing and release program. Adults used in diet studies were collected from pupation bins so that there was no variation in beetle age.

### 2.3. Utility of Artificial Diet for Larvae

In this study, first instars were placed on artificial diet to determine if it could be an alternative to fresh leaves. Commercially available Colorado potato beetle diet mix was used as an artificial diet for overwintering beetles (Frontier Agriculture Sciences Product #F9380B). The diet preparation instructions were followed and 3.5 g of whole fresh air potato leaves were added to the ingredients in a blender. The diet was blended for one minute and poured in long, roughly 2.5 cm-wide, strips across wax paper. The diet cooled and hardened for 20 min and the wax paper was cut into 2.5 cm square sections and frozen in a clean, sealed tub.

In a 4-L ventilated plastic bin, 17–20 first instar larvae were placed on either artificial diet or leaves (replicate 1, n = 20; replicate 2, n = 17; replicate 3, n = 17). They were checked for mortality after 18 and 24 h. Larvae within each replicate emerged on the same day. All replicates were kept at 27 ± 2 °C, 65 ± 10% RH, and 14L:10D.

### 2.4. Adult Starvation Trials

Beetles from larvae raised on air potato leaves were placed in a terrarium (33 cm × 18.5 cm × 21 cm) with autoclaved sand as a substrate. A total of four trials were run at different times for beetles fed leaves for 7 days (A, B) or newly emerged beetles not fed (C, D) air potato leaves prior to the study. Trials A and B were held in June and July of 2020 and trials C and D were held in October of 2020. Each trial included four replicates for each treatment: fresh leaf tissue only (“Leaves”), bulbils only (“Bulbils”), leaf and bulbil tissue (“Control”), or water only (“Starved”). In Trials A and B, 25 beetles were used per replicate; in Trials C and D, 12 beetles were used per replicate. These numbers were based on beetle availability. All replicates were kept at 27 ± 2 °C, 65 ± 10% RH, and 14L:10D. Water was provided in all treatments via wet cotton wicks. Mortality was recorded daily. Beetles used in trials A and B emerged within three days of each other and were fed leaf tissue for seven days prior to receiving a diet treatment. These trials were carried out over the course of a month. Beetles used for trials C and D emerged on the same day and received a diet treatment without being fed previously. These trials ended after 12 days as 100% of beetles in the “Starved” treatment had died.

### 2.5. Statistical Analysis

For artificial diet and starvation studies, statistical analysis was performed using software R [15]. Trials with the same experimental design were combined and a single analysis performed. Library Rmisc was used for basic summary statistics. The Shapiro–Wilk test for normality revealed that the data were not normally distributed (W = 0.73 *p* < 0.01). Therefore, for the artificial diet experiment, a Scheirer–Ray–Hare test was performed with a post hoc Dunn test with “treatment”, “time”, and their interaction as factors and “percent mortality” as the response variable. For the starvation trials, trials with beetles fed leaves before being given an experimental diet (A and B) had the same experimental design so the data were combined, and a single analysis performed. The same procedure was used for data from beetles that did not feed before receiving a diet (C and D). A Kruskal–Wallis Test was performed with a post hoc Dunn test with “treatment”, “trial”, and their interaction as factors and “percent mortality” as the response variable.

## 3. Results

### 3.1. Air Potato Propagation Resources

The air potato vine propagation and beetle rearing method can provide tens of thousands of beetles each year depending on the facility size, amount of tissue continuously provided to the beetles, and number of available rearing personnel. The facility currently has one 9.14 m by 15.24 m greenhouse, one 13.72 m by 7.62 m screenhouse, and an outdoor area of 0.1 ha with 180, 100, and 1300 pots of air potato, respectively. Cultivating plant tissue is the most space-consuming component, but it is vital to the propagation method. Our team includes four members, three of whom work full time on the air potato project at peak production periods. This can decrease to one full-time and one part-time team member in winter months. Team members perform research as well as rearing tasks.

### 3.2. Utility of Artificial Diet for Larvae

Slightly different numbers of beetles were used in each replicate thus results are reported as percentages. The Scheirer–Ray–Hare test showed no significant effect of “time” on percent mortality (H = 0.60 *p* = 0.44), and there was no significant interaction between “treatment and time” (H = 0.60 *p* = 0.44). A significant effect on percent mortality was noted when larvae were fed artificial diet (H = 9.62 *p* < 0.01). One trial with three replicates per treatment was deemed sufficient as results were well-defined. After 18 h, a mean of 79.9% of larvae provided with artificial diet were dead. Observations showed that larvae on diet left the diet strips likely in search of alternative food sources. After 24 h, 100% of larvae on artificial diet were dead and 100% of larvae fed fresh leaf tissue remained alive.

### 3.3. Adult Starvation Trials

Slightly higher overall mortality in trials B and D was likely because they were conducted later in the year when beetles generally begin to experience higher mortality. However, there were no significant differences between trials A and B (Fed) or trials C and D (Not fed), so the data were averaged and reanalyzed together. For trials A and B, data did not pass the test for normality (Shapiro–Wilk W = 0.76 *p* < 0.01). The Kruskal–Wallis Test showed a significant effect of the treatment on percent mortality of beetles (H = 25.23, *p* < 0.01) (Figure 1). For trials C and D, data did not pass the test for normality (Shapiro–Wilk W = 0.81 *p* < 0.01). The Kruskal–Wallis Test showed there was a significant effect of the treatment on percent mortality of beetles (H = 25.74 *p* value < 0.01) (Figure 1).

### 3.4. Lilioceris cheni Production and Releases

From 2014 to 2021 the FDACS-DPI facility produced 407,554 beetles and released 366,384 of them at nearly 7000 sites across Florida, Georgia, Mississippi, Alabama, Louisiana, and Texas (Table 1). Beetle quality was assessed through longevity studies that show a median lifespan equivalent to that of colonies assessed in 2017 (Appendix A) [13]. In the 2019–2020 funding cycle, the cost of beetle production was USD 3.17 per beetle. In the following cycle, 2020–2021 costs were reduced to USD 2.17 per beetle, and in the current cycle further reduced to USD 1.82 per beetle. Costs included labor of a Biological Scientist II and two OPS Laboratory Technician IVs in the first two cycles but only the Laboratory Technician IVs in the third cycle. Costs also included materials in the equipment and supply list (Appendix A) purchased as needed and shipping beetles which is expensive. The FDACS-DPI procedure was followed for shipping and distribution [14]. The majority of these releases were performed by residents and scientists who received beetles via mail.

## 4. Discussion

The FDACS-DPI air potato beetle mass-rearing protocol has been very successful for producing and distributing hundreds of thousands of beetles. The mass rearing process at the level presented here requires a minimum of basic supplies and equipment [14] and a facility with considerable indoor or outdoor rearing space. The size of the facility depends on the desired level of production. Cultivating plant tissue is the most space-consuming component, but it is vital to the method. The limiting factor in mass rearing systems for weed biological control is often the amount of plant tissue and plant health [16,17]. Use of artificial diet, antimicrobials, and quality control are vital for efficient and successful production [17,18].

The artificial diet developed was a successful overwintering food source for adults but not for larvae. This diet has supported successful overwintering of adults for eight consecutive years. Larvae quickly leave the diet in search of a suitable food source and many die within 18 h of being provided the artificial diet only. One hundred percent mortality occurs after 24 h. Thus, maintaining a small number of air potato plants through winter is necessary for the adult artificial diet and for feeding any larvae that may be produced. The diet allows for a large reduction in the number of vines needed and thus a reduction in space and personnel hours. An additional observation was made with adults fed artificial diet over the winter months. The Nepalese biotype beetles were observed mating, but never oviposited. Less than one week after being provided with fresh leaf tissue, oviposition resumed. This preliminary evidence suggests artificial diet cannot be used for adults in production, only for overwintering maintenance of the colony. However, several beetle species have been mass reared on artificial diet or combinations of diet and the natural food source [19,20]. Thus, it is possible alterations to the formula, or a combination of diet and fresh leaf tissue could support larval development.

Adult air potato beetles experience rapid and significant mortality when starved immediately after emergence. If fed after emergence but starved at a later point they experience significant mortality but at a slower rate. Adults also experience higher mortality when fed bulbils as compared to leaves or leaves and bulbils. This information shows that it is feasible to occasionally starve adults for short periods such as over weekends or during shipping. However, adults should always be fed fresh air potato leaves upon emergence to reduce mortality. Additionally, adults should be shipped with some air potato leaf tissue when the recipients are releasing on an unconfirmed plant source. If the source is a plant other than air potato, which may happen when shipping to non-scientist requestors or reporters, the beetles should have enough dietary resources to relocate if indeed there is air potato vine within their flight range, which has been estimated to be between 1.4 and 8 km per year [21,22].

Several approaches can be taken for disseminating agents depending on the goal of the program. Each of these approaches works best when specific release instructions are included with the beetles (Appendix A). It is highly recommended beetles are distributed in groups of 50–100, accounting for the level of vine infestation and area of coverage. According to one study, future releases of 100 adults spaced several kilometers apart could promote rapid colonization [21]. The authors support large releases as the field mortality is somewhat unknown.

Distribution and establishment of the agent was the ultimate goal of this mass-rearing operation. This agent has been disseminated across Florida for nearly a decade; however, after adverse weather events such as hurricanes, flooding, summer heat waves, and spring drought, beetle populations can be drastically reduced. Therefore, additional releases must be made in this region to support populations in recovery. Additionally, beetles released in the southern half of Florida have shown poor establishment, had lower reproduction, and caused less damage to vines than by beetles further north (Kraus unpublished data). Further research on the lack of beetle establishment in the southern region of Florida is warranted. Annual releases may be necessary in southern Florida, which would support continuation of the mass-rearing program.

## Figures and Tables

**Figure 1 insects-13-00065-f001:**
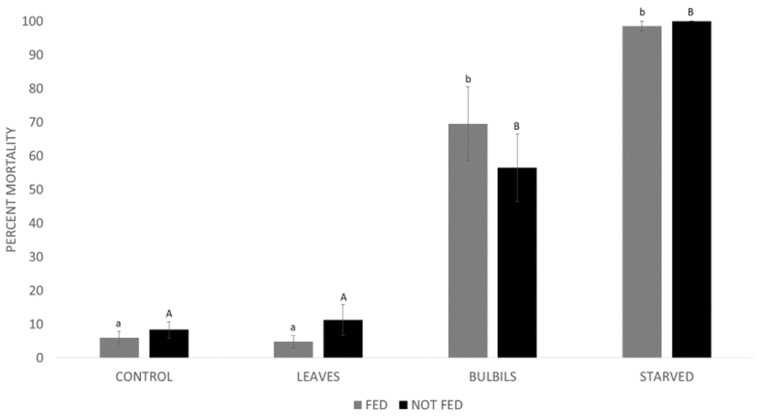
Percent mortality of air potato beetles on four diets after a one week feeding period (FED) on air potato leaves or testing at emergence (NOT FED). Letters indicate statistically significant differences in the two separate analyses. Results show mortality at the end of each experiment which was one month (FED) or 12 days (NOT FED).

**Table 1 insects-13-00065-t001:** The number of the two biotypes of beetles, Chinese and Nepalese, produced and released annually by the Gainesville facility. The grand totals of beetles produced and released from 2014–2021 include the number of sites where beetles were released in Florida and five other states. The asterisks in the “Release Sites” column denote the following: In 2014–2015 the number of counties in Florida is reported rather than the total number of release sites. Beetles were only released in Florida in this cycle. In 2017–2018 only the number of sites in Florida are reported although there were additional release sites in four other states. The grand total is thus slightly underreported due to these missing data.

Funding Cycle	Produced			Released			Release Sites
	Chinese	Nepalese	Total	Chinese	Nepalese	Total	Total
2014–2015	61,950	2686	64,636	56,702	83	56,785	26 *
2015–2016	48,540	1280	49,820	45,690	300	45,990	826
2016–2017	44,756	7169	51,925	39,655	5145	44,800	950
2017–2018	58,978	7150	66,128	56,482	6870	63,352	1377 *
2018–2019	58,000	510	58,510	53,298	0	53,298	1444
2019–2020	49,306	764	50,070	45,482	0	45,482	1387
2020–2021	56,515	9950	66,465	50,497	6180	56,677	971
Grand Totals	378,045	29,509	407,554	347,806	18,578	366,384	6981 *

## Data Availability

Data is available upon request through the Florida Department of Agriculture and Consumer Services. https://fdacs.mycusthelp.com/WEBAPP/_rs/(S(ptvoo05yhfsyz0hlmltsu4og))/supporthome.aspx (accessed 28 December 2021).

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
