# Peer review of "Advancements in Mass Rearing the Air Potato Beetle *Lilioceris cheni"

_insects, 2022, doi:10.3390/insects13010065_

Round 1

Reviewer 1 Report

This manuscript is more a report about the rearing program of the beetle than a scientific publication. Tests presented in this report are not scientific sound to be considered in a scientific publication.

The author could change the article type, it's up to the authors and editors.

Author Response

Thank you for taking the time to review our article.

Reviewers comment: "This manuscript is more a report about the rearing program of the beetle than a scientific publication. "

Author Response: The authors agree that the paper largely reported on the mass rearing program of the beetles and are altering the content according to the editors recommendations. The article has been shortened to reduce the details on the mass rearing aspect and focuses on the experiments which reveal potential to reduce the amount of air potato leaf tissue used.

Reviewer Comment: "Tests presented in this report are not scientific sound to be considered in a scientific publication."

Author Response: The authors believe the experiments included in this article are sound and appropriate for peer reviewed publication. We have used an appropriate experimental design, statistical analysis where appropriate, and reported on the significance of these findings. We concede our experiments were not highly technical but maintain they contribute to the literature on mass rearing biological control agents.

Reviewer comment: "The author could change the article type, it's up to the authors and editors."

Author Response: The authors would like to maintain the current article type. We believe our reporting on the mass rearing process and advancements we have made on the system are appropriate for this special issue.

Detailed revisions which address this reviewers comments are provided in the response to the editors and resubmitted manuscript . This was done as the reviewers comments are broad and their concerns are addressed in the major revisions requested by the editors.

Reviewer 2 Report

I found the manuscript "A Mass Rearing Method for the Air Potato Beetle Lilioceris cheni" well written and full of interesting data. For this reason I recommend the publication, with just few editing changes.

Author Response

The authors thank you for taking time to review this article. 

Reviewers comment: "I found the manuscript "A Mass Rearing Method for the Air Potato Beetle Lilioceris cheni" well written and full of interesting data. For this reason I recommend the publication, with just few editing changes."

We are working on the major revisions suggested by other reviewers and editors and hope that any minor editing changes will be corrected in the process.

Reviewer 3 Report

Manuscript “Insects-1385467, A Mass Rearing Method for the Air Potato Beetle Lilioceris cheni” reports on detailed methods and authors’ experience on insectary rearing of an important weed biological control agent in Florida, USA. Information will assist those attempting to propagate and distribute the beetle for biocontrol of air potato vine. The manuscript is well written and flows well from start to end with an excellent discussion. I recommend consideration in the special issue on “Rearing Techniques for Biocontrol Agents of Insects, Mites, and Weeds” with minor revision. I added some editorial notes and suggestions in the attached PDF for authors amendments to a revised manuscript. Some of my notes are:

  • As written, I think that Material and Methods represents an anecdotal account of the authors’ experience with mass rearing of this beetle. It would be great to make M&M into segments on the topics linked to sections in results: describing the food plants, beetle rearing and the other diet experiments, control of major arthropod pests that can disrupt rearing activities, control of insect pathogens, shipping procedures, quality control performing experiment to test longevity and starvation, and statistical analysis.
  • Please add a cost figure per rearing a thousand beetles.
  • Results can be enhanced with a paragraph on quality control and how to control bacterial and fungal diseases in a mass rearing situation where hundreds of beetles feed and drop frass on leaves in rearing containers.  
  • Any reported natural enemies of Lilioceris cheni in the release areas?
  • Where and how many localities for release in Florida? Important to add a column by total release in table 1 for number of lots and localities each year.
  • Percent mortality of air potato beetles after 18 hours and 24 hours on artificial diet (Figure 2). The three replicates should be lumped to provide bars with means and SEM for the 18- hour and 24-hour observations.
  • Appendix A and B may be provided as supplementary files but should not be considered a part of the manuscript body. The photos for cotton wicks and plant rearing containers, with and without the bamboo sticks are not essential for readers’ understanding the rearing methods.  

Author Response

The authors would like to thank the reviewer for their time and effort on this extensive review. The comments and constructive criticism will greatly improve the article. A point by point response to overall comments is detailed below. Responses to comments within the PDF provided are attached.

Additional changes in the manuscript are included in the track changes in the response to the editors as major revisions were requested.

Reviewers Comment: "As written, I think that Material and Methods represents an anecdotal account of the authors’ experience with mass rearing of this beetle. It would be great to make M&M into segments on the topics linked to sections in results: describing the food plants, beetle rearing and the other diet experiments, control of major arthropod pests that can disrupt rearing activities, control of insect pathogens, shipping procedures, quality control performing experiment to test longevity and starvation, and statistical analysis."

Author Response: The authors agree subsections will improve readability of this section. These have been added to the sections that were retained.

Reviewer Comment: "Please add a cost figure per rearing a thousand beetles."

Author Response:  A cost figure has been added for the two most recent funding cycles in the results section. Although this section may be removed by editors preference.

Reviewer Comment: "Results can be enhanced with a paragraph on quality control and how to control bacterial and fungal diseases in a mass rearing situation where hundreds of beetles feed and drop frass on leaves in rearing containers."

Author Response:  The authors agree they could comment on quality control. Several sentences have been added to the methods section to address this.

Reviewers comment: "Any reported natural enemies of Lilioceris cheni in the release areas?"

Author Response: Wasps have been observed consuming larvae in release areas. Beetle wings have been observed in lizard feces in the production areas and it is assumed some mortality occurs in the field due to these predators. A statement reflecting on these observations has been added to the discussion.

Reviewers Comment: "Where and how many localities for release in Florida? Important to add a column by total release in table 1 for number of lots and localities each year."

Author Response: A column has been added to table 1 to denote the total number of release sites where data was available. 

Reviewers Comment: "Percent mortality of air potato beetles after 18 hours and 24 hours on artificial diet (Figure 2). The three replicates should be lumped to provide bars with means and SEM for the 18- hour and 24-hour observations."

Author Response: Statistical analysis was performed using the treatment (artificial diet or leaves) and time (18 or 24 hours). The figure has been generated as suggested. However, standard error is extremely low and cannot well visualized.

Reviewers comment: "Appendix A and B may be provided as supplementary files but should not be considered a part of the manuscript body. The photos for cotton wicks and plant rearing containers, with and without the bamboo sticks are not essential for readers’ understanding the rearing methods."

Authors Response: The authors agree this information is suitable for supplemental material. The appendices have been relabeled as such.

Round 2

Reviewer 1 Report

I reassert my previous opinion:

Table 1 is a compilation of beetle's production in the facility. Fine for a report but this data do not suport any hypothesis, as any scientific experiment should have.

Figure 1 coulod be deleted and just add one simple sentence in the Results: "artificial diet does not work since 100% mortality was achived in 24 hours".

Figures 2 & 3 show results of a bad experiment, since authors stopped evaluating survival when all beetles died in the control. What is the point of such experiment?? I do not see...

Author Response

Reviewer comment: Table 1 is a compilation of beetle's production in the facility. Fine for a report but this data do not suport any hypothesis, as any scientific experiment should have.

Authors Response: Table 1 which shows the production and distribution of the beetles is vital to the paper. It is representative of the efficacy of the rearing method.

Reviewer comment: Figure 1 coulod be deleted and just add one simple sentence in the Results: "artificial diet does not work since 100% mortality was achived in 24 hours".

Authors Response: The figure may be removed as there is no significant effect of 'time' or interaction between 'treatments and time'. It cannot be replaced by such an utterly simplistic statement. The authors have clearly stated the results.

Reviewer comment: Figures 2 & 3 show results of a bad experiment, since authors stopped evaluating survival when all beetles died in the control. What is the point of such experiment?? I do not see...

Authors Response: It is grossly apparent the reviewer has not understood the experiment as it did not end when the beetles died in the control group. Approximately 80-90% of beetles in the control survived to the end of the experiments. It is a well-designed, scientifically sound experiment which shows that beetles will not survive longer than 2 weeks without a food source. It also shows that the beetles will experience high mortality on a bulbil only diet. The point of such an experiment was to provide evidence for how long beetles can survive without leaf tissue.